# Stimulation-induced cytokine polyfunctionality as a dynamic concept

Kevin Portmann, Aline Linder, Klaus Eyer*[†]

Laboratory for Functional Immune Repertoire Analysis, Institute of Pharmaceutical Sciences, Department of Chemistry and Applied Biosciences, ETH Zurich, Zurich, Switzerland

*For correspondence:
eyerk@biomed.au.dk

Present address: [†]Department of Biomedicine, Aarhus University, The Skou Building Høegh-Guldbergs Gade 10, DK-8000, Aarhus, Denmark

Competing interest: The authors declare that no competing interests exist.

**Abstract** Cytokine polyfunctionality is a well-established concept in immune cells, especially T cells, and their ability to concurrently produce multiple cytokines has been associated with better immunological disease control and subsequent effectiveness during infection and disease. To date, only little is known about the secretion dynamics of those cells, masked by the widespread deployment of mainly time-integrated endpoint measurement techniques that do not easily differentiate between concurrent and sequential secretion. Here, we employed a single-cell microfluidic platform capable of resolving the secretion dynamics of individual PBMCs. To study the dynamics of poly-cytokine secretion, as well as the dynamics of concurrent and sequential polyfunctionality, we analyzed the response at different time points after ex vivo activation. First, we observed the simultaneous secretion of cytokines over the measurement time for most stimulants in a subpopulation of cells only. Second, polyfunctionality generally decreased with prolonged stimulation times and revealed no correlation with the concentration of secreted cytokines in response to stimulation. However, we observed a general trend towards higher cytokine secretion in polyfunctional cells, with their secretion dynamics being distinctly different from mono-cytokine-secreting cells. This study provided insights into the distinct secretion behavior of heterogenous cell populations after stimulation with well-described agents and such a system could provide a better understanding of various immune dynamics in therapy and disease.

## eLife assessment

This **useful** study uses a microfluidic method to evaluate the ability of single human white blood cells to produce combinations of cytokines and the evidence that this takes place is **solid**. The paper highlights polyfunctionality using data that are similar to a prior dataset from the same group. The authors comment that, in analysis of larger panels, single cells rarely make more than 2 or 3 cytokines so that investigation of 3 cytokines at a time is sufficient to investigate this phenomenon. Coupling this approach to other modes of single cell analysis may provide greater insight into what limits simultaneous production of multiple cytokines.

## Introduction

Cytokines are small proteins that act as key immune messengers. Secreted by various cell types, they enable coordination and communication and play a fundamental role in mediating and regulating immune responses in infection and disease (*Banyer et al., 2000*; *Zhang and An, 2007*). Each cytokine has one or more defined functions in orchestrating pro- and anti-inflammatory responses. While traditional research was focused on mono-cytokine-secreting cells to understand the functions of individual cytokines, recent observations on the single-cell level introduced the concept of polyfunctionality (*Perfetto et al., 2004*; *Boyd et al., 2015*). Polyfunctional cells are characterized by the ability

to secrete multiple cytokines simultaneously, communicating a combination of different information to their microenvironment. This feature is believed to be crucial in shaping specific and targeted immune responses, allowing for a more diverse and adaptive defense against pathogens and diseases (*Boyd et al., 2015*; *Minton, 2014*; *Foley, 2012*).

While polyfunctionality can occur in various immune cell types, the phenomenon has been extensively studied for T cells due to their central and plastic role in orchestrating adaptive immune responses (*Foley, 2012*; *Levinson et al., 2020*). T-cell polyfunctionality has been identified in both CD4+ and CD8+ subsets, with cells being capable of secreting combinations of cytokines. These include interferon-gamma (IFN-γ), tumor necrosis factor-alpha (TNF-α), and interleukin-2 (IL-2) (*Lam et al., 2018*). The presence of polyfunctional T cells has also been associated with more effective control of chronic microbial infections, including human immunodeficiency virus (HIV), cytomegalovirus (CMV), and hepatitis c virus (HCV) (*Casazza et al., 2006*; *Ciuffreda et al., 2008*; *Duvall et al., 2008*), and increased, beneficial responses to vaccination (*Lindenstrøm et al., 2009*; *Precopio et al., 2007*). These beneficial effects were accredited to enhanced T cell effector functions and the improved ability to coordinate and modulate immune responses. Enhanced tumor control has also been related to polyfunctional T cells (*Ding et al., 2012*; *Imai et al., 2020*), which has been leveraged by the introduction of chimeric antigen receptor (CAR) T cell therapy. Indeed, an exploratory study showed that the polyfunctional profile of CAR T cells was significantly associated with a better clinical outcome during treatment (*Rossi et al., 2018*). Therefore, polyfunctional cells play an essential role in immunity and immunological responses.

However, polyfunctionality as a concept is not simply beneficial in all cases, since in some innate immune cells it has also been linked to age-related functional and cognitive decline in elderly adults (*de Pablo-Bernal et al., 2016*). This ambivalence is not surprising considering the central role of polyfunctional immune cells in communication and initiation of specific immune responses (*Minton, 2014*). Therefore, a thorough and deep-phenotypic characterization of polyfunctionality is needed to further understand and link potential clinical outcomes to individual polyfunctional responses.

Given the importance of polyfunctional cells in fine-tuning immune responses and the potential of a more comprehensive view of immune activation that their analysis offers, our current understanding remains limited to the application of endpoint and integrative measurements. However, polyfunctionality demands the analysis on the single-secreting cell level and cannot be measured in the supernatant of secreting cells. Most common measurement methods include multiparametric ELISpot (where spots overlap), Isoplexis, and multicolor intracellular cytokine staining (where the cells are analyzed individually for several cytokines) (*Perfetto et al., 2004*; *Gazagne et al., 2003*; *Ma et al., 2011*). While all allow correlation on the single-cell level, dynamic resolution of secretion is not provided as these methods are endpoint measurements. Therefore, potential differences in simultaneous or consecutive cytokine secretion are not revealed as they are best resolved dynamically. The dynamic aspect of polyfunctionality often falls short and only a few studies have so far reported dynamic measurements of cytokine secretion (*Han et al., 2012*; *Portmann et al., 2023*), which remains crucial to understand whether functionality adapts over time or is communicated simultaneously. Lastly, as the function of many cytokines is linked to their location, such as membrane-bound, shed, or secreted species (*Elliott and Sutterwala, 2016*; *Gerspach et al., 2000*), a direct functional analysis of the secreted protein level might be beneficial. Therefore, new methodological advances are needed to provide such a resolution.

To overcome these limitations, we recently developed a dynamic multiplexed cytokine secretion platform using microfluidic single-cell encapsulation (*Figure 1A*; *Portmann et al., 2023*). In brief, the platform enabled highly sensitive measurements of individual cytokine-secreting cells (CSCs) for multiple cytokines, and most importantly, provided dynamic resolution into their secretion behavior. In short, in this workflow, the cells are encapsulated into water/oil emulsions (droplets), immobilized, and their secretion quantified over time. In every droplet, three sandwich immune assays were run in parallel, allowing the simultaneous detection of up to three secreted cytokines (*Figure 1B–D*). Through quantitative and longitudinal measurements, the analysis of multiplexed cytokine secretion enabled a dynamic examination of secretion. In the study described herein, we used this platform to investigate the differential polyfunctional cell responses in human peripheral blood mononuclear cells (PBMCs) following stimulation with established activation cocktails over different stimulation times. By doing so, we gained insights into the occurrence and disappearance of polyfunctional cells and

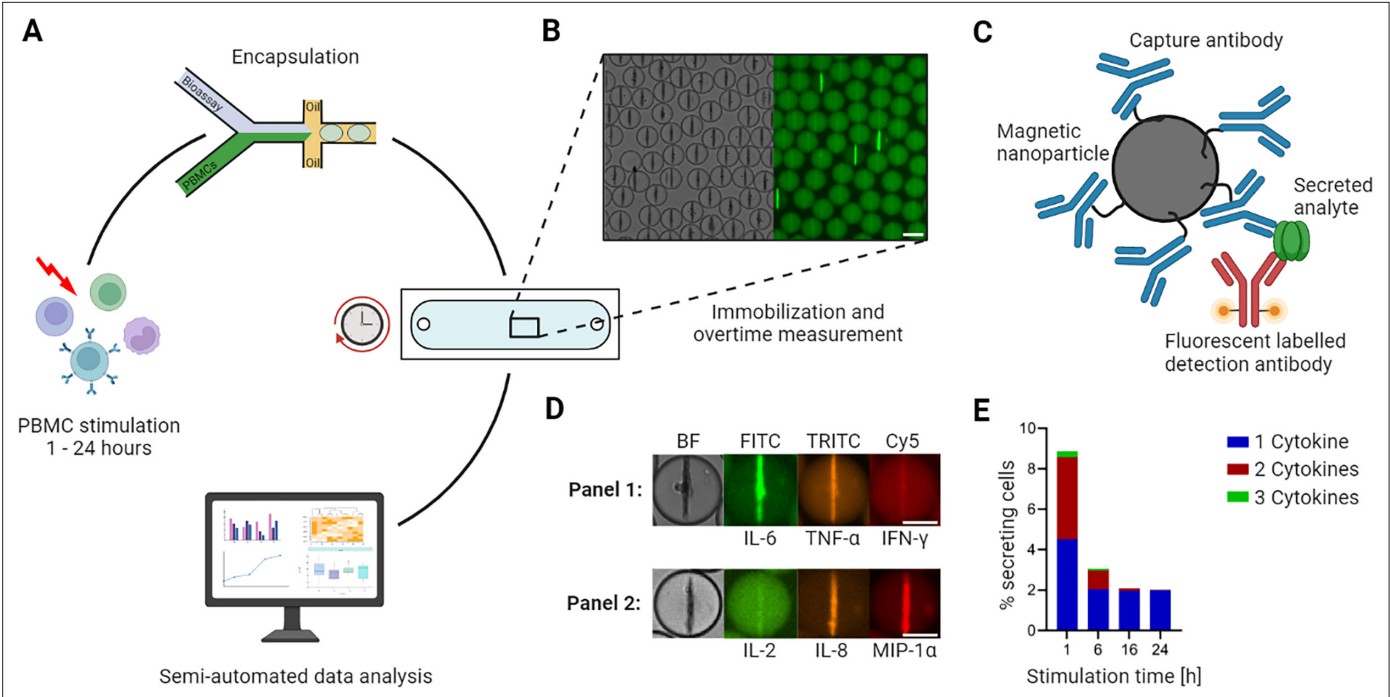

**Figure 1.** Workflow of the microfluidic cytokine secretion measurements. (**A**) The experimental protocol involved the stimulation of peripheral blood mononuclear cells (PBMCs) in bulk, followed by their encapsulation with assay reagents in 65 picolitre water/oil emulsions (called droplets). Subsequently, the droplets containing cells were immobilized in an observation chamber and imaged over 4 hr every 30 min, followed by a semi-automated data analysis pipeline. (**B**) Micrographs of an array of droplets immobilized in the observation chamber. The insert shows the droplets in brightfield and one fluorescence channel. Droplets with lines indicate the presence of a cytokine-secreting cell in this container. Scale bar: 50 µm. (**C**) Assay principle to measure cytokine secretion. Assay reagents consist of 300 nm in diameter paramagnetic nanoparticles and fluorescently labeled detection antibodies. The nanoparticles are functionalized with capture antibodies specific to the cytokines of interest. When secreted, the cytokine binds to the capture antibody with subsequent relocation of one particular fluorescently labeled detection antibody. The application of a magnetic field aligns the nanoparticles into an elongated aggregate, making it possible to measure fluorescence relocation for every channel and to measure fluorescence relocation onto the nanoparticles (as seen in B). (**D**) Nanoparticles functionalized against different cytokines allow multiplexing for up to three cytokines. The images shown represent exemplary cells secreting IL-6$^+$/TNF-$\alpha^+$ (in a panel measuring IL-6/TNF-$\alpha$/IFN-$\gamma$) and IL-8$^+$/MIP-1$\alpha^+$ (IL-2/IL-8/MIP-1$\alpha$). Scale bars: 25 µm. (**E**) Response of LPS-stimulated PBMCs for various stimulation times, namely 1, 6, 16, and 24 hr. The resulting percentage of secreting cells was binned for polyfunctionality, i.e., cells secreting one (blue), two (red), or all three (green) measured cytokines (panel IL-6/TNF-$\alpha$/IL-1$\beta$). Panels A and C created with BioRender.com, and published using a CC BY-NC-ND license with permission.

The online version of this article includes the following figure supplement(s) for figure 1:

**Figure supplement 1.** Cytokine supernatant concentrations after 24 hr in response to various stimulants.

differentiated simultaneous from consecutive secretion in response to various stimuli, providing a deeper understanding of their functional dynamics.

## Results

### Cytokine polyfunctionality of stimulated PBMCs changes with prolonged LPS stimulation

First, we aimed to investigate the dynamic nature of polyfunctionality. Therefore, we stimulated PBMCS with LPS and analyzed their cytokine response at various times after initial stimulation, resembling a potential in vivo situation in which immune responses are mounted and developed over time (*Tawfik et al., 2020*). The secretion response of IL-1$\beta$, IL-6, and TNF-$\alpha$ was measured after different time points after stimulation (*Figure 1E*), and we binned the observed frequency of CSCs into mono-, bi-, and tri-secreting cells, with the latter two categories representing polyfunctional cells. Early after stimulation, 4.5% of the PBMCs were secreting one, and 4.4% were secreting two or more of the

assayed cytokines. Interestingly, a clear decrease in polyfunctionality of the secreted cytokines was observed with prolonged stimulation times, and all polyfunctional cells disappeared after 24 hr of stimulation. In contrast, the population of mono-secretion was only reduced by roughly 50% in the first 6 hr, and remained constant with prolonged stimulation (from 4.5 to 2% for the 1- and 24 hr stimulations, respectively). Overall, polyfunctionality in this experiment was only short-lived, with an over-proportional decrease in polyfunctionality not explained by the overall reduction of CSCs suggesting other mechanisms at play.

## Polyfunctionality dynamics are largely panel-independent, but various stimulations show different evolutions of polyfunctionality

Next, we wanted to characterize whether this observation was panel or stimulant-specific since LPS mainly activates through direct TLR4 signaling (*Lu et al., 2008*). This is why we decided to expand the used stimulants to well-known standards for in vivo activation of immune cells (*Finco et al., 2014*), including zymosan, phorbol myristate acetate (PMA)/ionomycin, anti-CD3/anti-CD28 antibodies, phytohemagglutinin (PHA) and RPMI only as the negative control. To identify cytokines of interest, the supernatant response to these six conditions after 24 hr of stimulation was examined (*Figure 1—figure supplement 1*). For this, the highly multiplexed cytokine storm CodePlex (Isoplexis) was used, being capable of the simultaneous detection of up to 18 cytokines. In response to our selection of

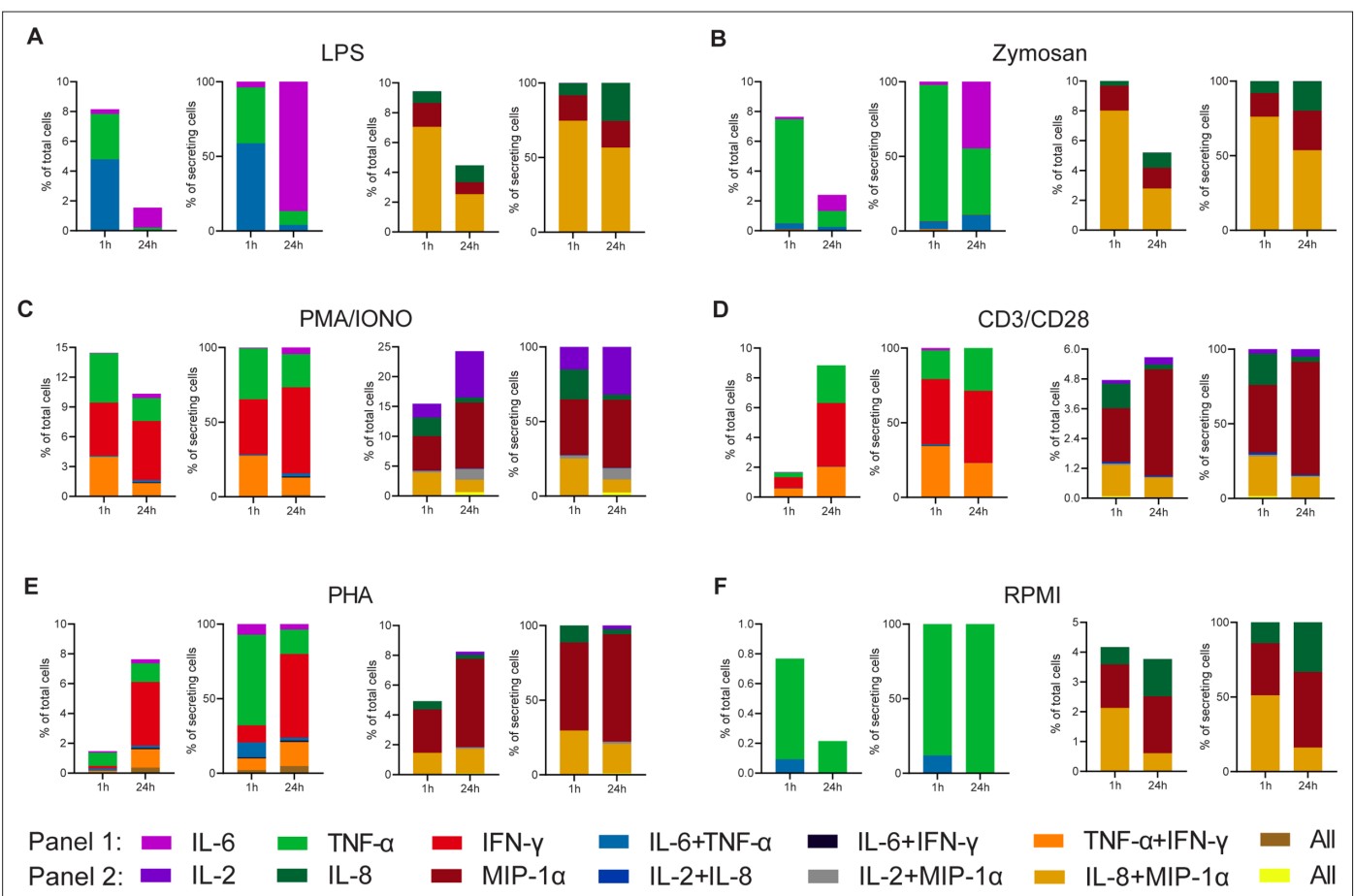

**Figure 2.** Cytokine polyfunctionality of stimulated peripheral blood mononuclear cells (PBMCs) changes with prolonged stimulation. PBMCs were stimulated with either 1 μg/ml LPS (**A**), 100 μg/ml zymosan (**B**), 50 ng/ml PMA + 1 μg/ml ionomycin (**C**), 5 μg/ml anti-CD3 (OKT3)/anti-CD28 (CD28.2, **D**), 10 μg/ml PHA-L (**E**), or media alone (**F**) for 1 and 24 hr and subsequently measured over 4 hr. Endpoint percentage cytokine-secreting cells (CSCs) were categorized into the following bins depending on the detected cytokines per droplet and the detection panel used IL-6 (violet), TNF-α (green), IFN-γ (red), IL-6⁺/TNF-α⁺ (blue), IL-6⁺/IFN-γ⁺ (black), TNF-α⁺/IFN-γ⁺ (orange), all three cytokines (brown) and IL-2 (dark violet), IL-8 (dark green), MIP-1α (dark red), IL-2⁺/IL-8⁺ (dark blue), IL-2⁺/MIP-1α⁺ (gray), IL-8⁺/MIP-1α⁺, or all three cytokines (yellow). The number of secreting cells in each bin is displayed as frequency of total measured cells (left panels) and frequencies of all CSCs per measurement (right panels). n_biological replicates=1.

stimulants, mainly GM-CSF, IFN-γ, IL-2, IL-6, IL-8, MIP-1α, MIP-1β, and TNF-α were detected in sufficiently high concentrations to assure secretion during the 24 hr stimulations. Since MIP-1β did not display large differences in measured concentrations in response to different stimulations, we decided to focus on IL-2, IL-6, IL-8, TNF-α, IFN-γ, and MIP-1α secretion for all subsequent measurements. These cytokines were measured in two panels of three cytokines each (IL-6/TNF-α/IFN-γ, and IL-2/IL-8/MIP-1α, also referred to as panels 1 and 2, *Figure 1D*). Short (1 hr) and long (24 hr) stimulation times were chosen in order to visualize dynamic changes and the influence of prolonged stimulation.

*Figure 2* shows the absolute (CSCs in all assayed PBMCs) and normalized (to all detected CSCs) frequencies for all tested cytokines, stimulants, and stimulation durations. The frequencies were binned into the respective polyfunctional population, depending on which cytokine was observed during the measurement. For LPS stimulations with the modified first panel (IFN-γ instead of IL-1β), the total number of CSCs decreased with prolonged stimulations from 8.2% and 9.4 to 1.6% and 4.5%, for the first and second panels, respectively (*Figure 2A*). These cells were found to only secrete TNF-α and IL-6, or IL-8 and MIP-1α, including combinations thereof. After prolonged incubation, i.e., 24 hr later, the double-positive IL-6 and TNF-α completely disappeared. A similar observation was also made for the other panel and the absolute decrease for both panels can be attributed to the reduced frequency of polyfunctional cells secreting either IL-6$^+$/TNF-α$^+$ or IL-8$^+$/MIP-1α$^+$, with a decrease of 4.7% and 4.5% for the two populations, respectively. While cytokine secretion decreased, it is important to note that prolonged stimulation times also resulted in higher frequencies of certain mono-secreting cell populations, as visible for IL-6 and IL-8. Therefore, the trend towards reduced polyfunctional cells after prolonged stimulation times does not necessarily correlate with the trends seen in the mono-secreting cells, and was not limited to the first panel tested. To visualize these different trends better, we normalized the percentage of all secretion categories to the total number of secreting cells. Indeed, we observed a shift to a pure mono-cytokine response after 24 hr, where mainly IL-6 secreting cells were detected. Interestingly, the relative frequency of only IL-8 secreting cells also increased, but only marginally when compared to the IL-6-secreting cells (threefold vs. 22-fold). Therefore, the disappearance of polyfunctional cells at prolonged incubation times was not only observed in both panels but was also not correlated directly with the appearance or disappearance of mono-cytokine-secreting cells.

We next wondered whether these observations were linked to the used LPS stimulation and, therefore, we looked into other stimulants. Similar to LPS, stimulation with zymosan led to a reduction of overall CSCs at prolonged incubation times (*Figure 2B*), with an overall reduction in the number of secreting cells in both panels (–5.2% and –5.3% for panel 1 and 2, respectively). Contrary to LPS stimulation, there only frequencies of IL-8$^+$/MIP-1α$^+$ secreting cells were reduced, whereas IL-6$^+$/TNF-α$^+$ cells remained stable at a very low frequency (0.4% and 0.2% for 1 and 24 hr, respectively). There was also a considerable mono-TNF-α and IL-6 response after 24 hr of stimulation (1.1% for both). Overall, there was a shift towards a mono-cytokine response for both LPS and zymosan stimulations with a reduction in the frequency of polyfunctional cells with prolonged stimulation.

Broadening the stimulated cell populations, PMA/ionomycin was used to induce cytokine secretion (*Figure 2C*; *Mandala et al., 2021*). Extended stimulation with PMA/ionomycin resulted in an overall decrease in the frequency of IL-6, TNF-α, and IFN-γ mono secreting cells, while simultaneously increasing the total number IL-2, IL-8, and MIP-1α mono secreting cells (–4.2% and +7.9%, respectively). This increase was mainly due to increased MIP-1α and IL-2 secretion at later time points. Interestingly, and again in contrast with the mono-cytokine observations, we observed a reduction in polyfunctionality for the first panel, with TNF-α$^+$/IFN-γ$^+$ secreting cells decreasing by 2.8%. However, no reduction in polyfunctional cells was observed for the second panel, since a decrease of IL-8$^+$/MIP-1α$^+$ secreting cells (–1.8%), was counteracted by the appearance of IL-2$^+$/MIP-1α$^+$ and IL-2$^+$/IL-8$^+$/MIP-1α$^+$ secreting cells (+1.5% and +0.6%, respectively).

Since we observed the additional appearance of polyfunctional cell populations using PMA/ionomycin, which also activated T cells, we wondered whether T cell polyfunctional dynamics were distinctly different from the dynamic observed during LPS and zymosan stimulation. In consequence, we also stimulated PBMCs with anti-CD3/anti-CD28 antibodies. This yielded an overall increase in the percentage of secreting cells for both panels of 7.1% and 0.9%, respectively (*Figure 2D*), and for the first time, we observed increased polyfunctionality in the first panel over prolonged incubation, with TNF-α$^+$/IFN-γ$^+$ secreting cells rising by 1.4%. Since this increase was not mirrored relative to all CSCs

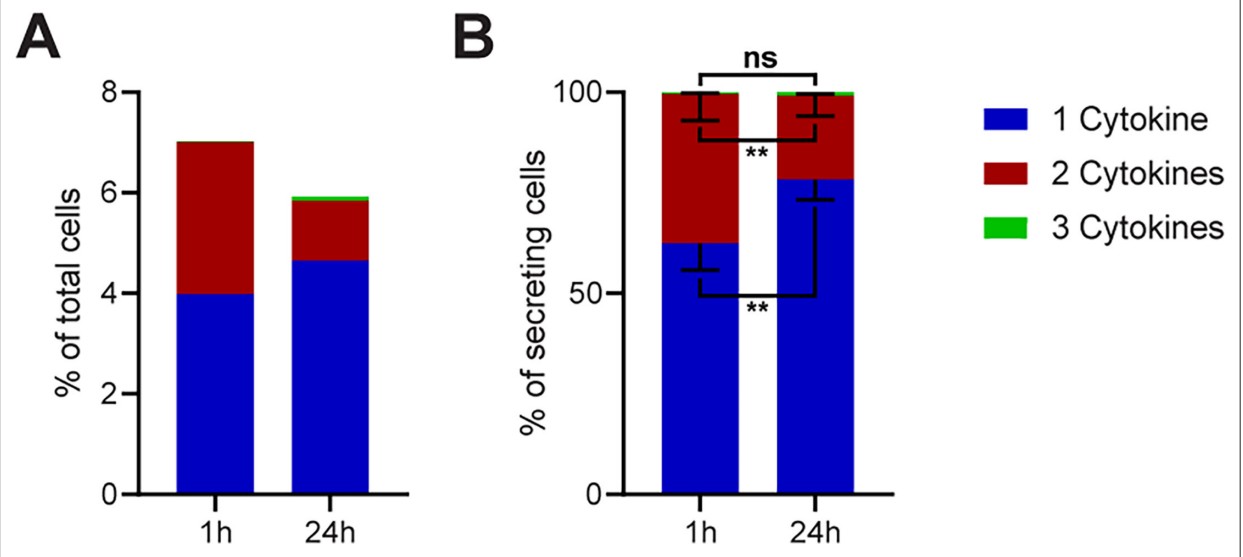

**Figure 3.** Polyfunctionality of all measured cytokines and stimulants (summary of *Figure 2*). All measured cytokines and stimulants were combined and binned into cells secreting one, two, or three cytokines simultaneously. (**A**) The average secreting cells in each bin over all the measurements in relation to all measured cells for 1- and 24 hr stimulations (see methods for details). (**B**) Average of the normalized secreting cells in each bin in relation to all secreting cells. Differences between 1- and 24 hr stimulations were assessed with paired two-sided t-tests with 95% confidence. Data depicted as mean ± SEM. n=12.

(–11%), it probably stemmed from the overall increase in CSCs. In contrary, there is a small reduction in the total frequency of IL-8+/MIP-1α+ secreting cells (–0.5%) and relative to all CSCs (–13%). Anti-CD3/anti-CD28 stimulation also was distinctly different from PMA/ionomycin, since we did not observe the appearance of IL-2+/MIP-1α+ or IL-2+/IL-8+/MIP-1α+ secreting cells. PHA stimulation led to a comparable cytokine response as anti-CD3/anti-CD28 (*Figure 2E*). The overall percentage increased for panels 1 and 2 from 1.5 to 7.6% and 4.9 to 8.2%, respectively. The was also an increase in TNF-α+/IFN-γ+ frequencies, which is comparable to the response after anti-CD3/anti-CD28 stimulation (+1.1%). In contrary, the frequency of IL-8+/MIP-1α+ secreting cells remained stable (+0.2%), while again a reduction was observed in relation to all CSCs (–9%).

To examine background secretion and the respective change in polyfunctionality, non-stimulated PBMCs were measured as well (*Figure 2F*). Non-stimulated cells also exhibited a decrease in overall CSCs for both panels (–0.6% and –0.4%, respectively) with the frequency of IL-8+/MIP-1α+ secreting cells decreased by 1.5%. It is worth mentioning that these changes are relatively small in comparison to stimulated cells.

Overall, there are mainly three polyfunctional cell populations for the measured cytokines and stimulants: IL-6+/TNF-α+, TNF-α+/IFN-γ+, and IL-8+/MIP-1α+. Whenever present in response to stimulation, these populations showed a subpopulation-specific change over time. For example, while IL-6+/TNF-α+ and IL-8+/MIP-1α+ secreting cells mainly disappeared for all stimuli, TNF-α+/IFN-γ+ increased in response to PHA and anti-CD3/anti-CD28, but not for the PMA/ionomycin stimulation. Therefore, it might be that the evolution of T cell polyfunctionality is stimulation-dependent and distinctly different from the dynamics observed after LPS and zymosan incubation.

## Prolonged incubation times significantly reduce the frequency of polyfunctional cells

To investigate if there is a significant difference in the frequencies of polyfunctional cells across all the measured conditions in *Figure 2*, we binned all stimulations into mono-, bi-, or tri-CSCs for 1- and 24 hr stimulations (*Figure 3*). Indeed, there was a global 2.3-fold reduction of cells secreting two or more cytokines after prolonged incubation times (from 3 to 1.3%), with a small concurrent increase in mono-CSCs (4 to 4.7%).

To visualize differences within the secreting cells, we normalized the data to all CSCs in every measurement (*Figure 3B*). Here, we observed a strongly significant increase in the relative number

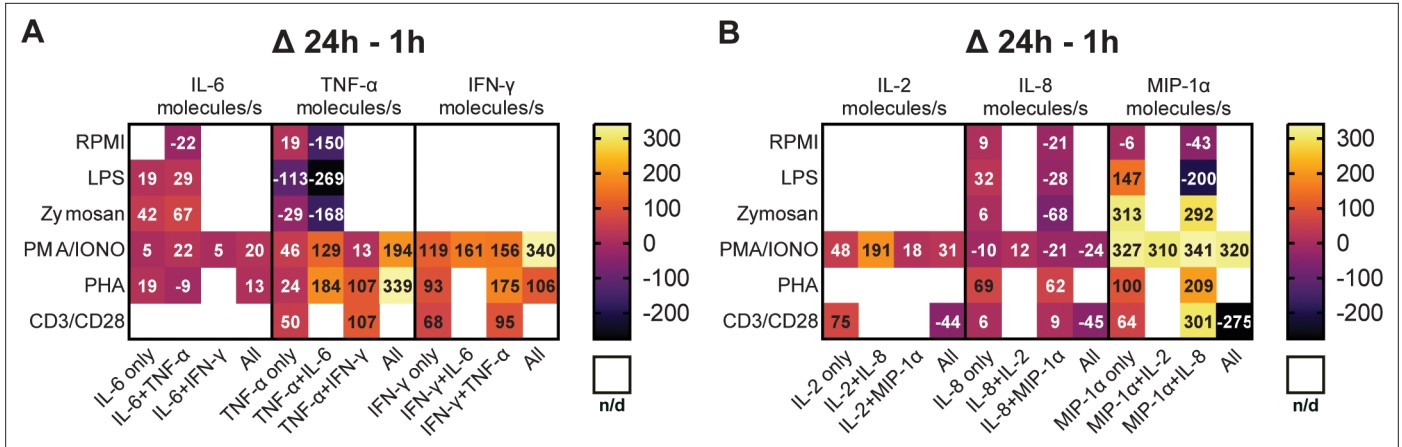

**Figure 4.** Secretion rates of polyfunctional cells vary for different cytokines and stimulation times. Normalized average secretion rates in molecules/s (1 hr subtracted from 24 hr measurement) over the measurement time (4 hr) for peripheral blood mononuclear cells (PBMCs) stimulated with LPS, zymosan, PMA/ionomycin, anti-CD3/anti-CD28, PHA-L, or media only for panel 1 (**A**) and panel 2 (**B**) Average secretion rates were extracted according to the different bins described in *Figure 2*. n/d=not detected.

The online version of this article includes the following figure supplement(s) for figure 4:

**Figure supplement 1.** Absolute secretion rates of polyfunctional cells vary for different cytokines and stimulation times.

of cells secreting only one cytokine (62.4 ± 6.3% vs. 78.3 ± 4.8%, p=0.006) and a significant decrease in the frequency of cells secreting two cytokines (37.2 ± 6.4% vs. 20.9 ± 4.8%, p=0.006). With triple-positive events being rare in our measurements, no significant difference was observed for cells secreting all three cytokines (0.4 ± 0.2% vs. 0.8 ± 0.4%, p=0.2). However, the disappearance of poly-functional CSCs at prolonged incubation times was strongly significant over all tested stimulants, although some stimuli and panels showed variations from this general trend (see *Figure 2*).

## The relationships between secretion rates and frequencies of polyfunctional cells are complex and cytokine-specific

Subsequently, we aimed to explore the potential correlation between the variations in polyfunctional CSCs and the levels of secreted cytokines. A hypothesis for the disappearance of polyfunctional CSCs might be a general reduction in cellular output, i.e., individual secretion rates. To study this, we calculated the average secretion rates across all conditions and cytokines during the measurement period and examined how these rates changed with prolonged stimulation times (*Figure 4*).

Looking first at the cells secreting IL-6⁺/TNF-α⁺ (*Figure 4A*), we observed a trend towards increased IL-6 secretion rates with prolonged stimulation times for LPS, zymosan, and PMA/ionomycin. Inter-estingly for TNF-α secretion, an increase in secretion rates was only observed for PMA/ionomycin (+129 molecules/s) and not for LPS and zymosan stimulations (–269 and –168 molecules/s, respec-tively). Notably, PHA stimulation yielded different results with only slightly decreased IL-6 secretion (–9 molecules/s) and increased TNF-α secretion (+184 molecules/s), considering the frequency of cells simultaneously secreting IL-6⁺/TNF-α⁺ stayed the same over time (0.2% for both time points). Of note is also that we compare absolute secretion values, relatively IL-6 secretion rates after PHA stim-ulation decreased by roughly 50% for IL-6⁺/TNF-α⁺ polyfunctional cells with prolonged stimulation. Compared to mono-cytokine-secreting cells, polyfunctional cells showed an increase of TNF-α secre-tion rates amongst all measured stimulations and with short and long incubation times (*Figure 4—figure supplement 1A, B*).

As described earlier, cells secreting TNF-α⁺/IFN-γ⁺ were observed after PMA/ionomycin, anti-CD3/anti-CD28, and PHA stimulations. The frequency of this population decreased for PMA/ionomycin stimulations but increased for anti-CD3/anti-CD28 and PHA stimulations with prolonged incubation (*Figure 2C/D/E*). Regarding secretion rates, a noticeable trend is observed for TNF-α, as the secreted amount increased across all three stimulations. Specifically, PHA and anti-CD3/anti-CD28 stimulations showed the biggest change, with both resulting in an increase of +107 molecules/s (*Figure 4A*). The same was observed for IFN-γ secretion rates, but here PHA and PMA/ionomycin showed the highest

increase (+175 and 156 molecules/s, respectively). Similar to IL-6+/TNF-α+, a trend towards higher IFN-γ and TNF-α secretion rates was observed in polyfunctional cells when compared to mono-CSCs (*Figure 4—figure supplement 1A and B*).

Cells simultaneously secreting IL-8+/MIP-1α+ were present in all the tested conditions (*Figure 2*) and their frequency decreased over time for all the use stimulants, with the exception of the PHA stimulation. In terms of secretion rate change with prolonged incubation for this subgroup, IL-8 secretion rates decreased for LPS, zymosan, PMA/ionomycin, and non-stimulated cells (−28, −68, −21, and −21 molecules/s, respectively), but increased for PHA and anti-CD3/anti-CD28 stimulated cells (+62 and +9 molecules/s, respectively, *Figure 4B*). MIP-1α secretion rates decreased for LPS and non-stimulated cells (−200 and −43 molecules/s, respectively), with higher secretion rates for all the other stimulants for IL-8+/MIP-1α+ secreting cells. While MIP-1α secretion rates generally seem to increase with prolonged stimulation, there is no overall trend for this subpopulation, with these cells seemingly being dependent on the stimulation and incubation time, which also does not correlate with observed frequency cells (*Figure 2*). However, similarly to the polyfunctional populations described above, there is a trend of higher cytokine secretion when comparing polyfunctional cells, to their mono-cytokine-secreting equivalents (*Figure 4—figure supplement 1C and D*).

PMA/ionomycin stimulation led to the appearance of IL-2+/MIP-1α+ secreting cells, not observed with all the other stimulants. Over the stimulation, there was a slight increase of IL-2 secretion in that population (+18 molecules/s), but a relatively high increase for MIP-1α (+310 molecules/s, *Figure 4B*). This potentially correlates with the overall frequency increase in these secreting cells (+1.5%, *Figure 2C*). Again, the polyfunctional cells secrete more cytokines when compared to their mono-cytokine-secreting equivalents (*Figure 4—figure supplement 1C and D*).

Overall, the polyfunctional response of PBMCs to well-known stimulants seemed highly complex. While the data displayed a trend towards fewer cells secreting more cytokines with prolonged stimulation, the secretion rates highly depended on the employed stimulus and measured cytokines, and the disappearance of polyfunctional cells could not be linked to lower secretion rates. However, increased secretion of polyfunctional cells, when compared to their mono-cytokine-secreting equivalents, was a consistent characteristic of polyfunctional cells.

## The secretion dynamics of polyfunctional cells revealed elevated secretion profiles for specific cell populations

Next, we wanted to investigate if the polyfunctional cells showed specific secretion dynamics and biological signatures, and whether they differed from their mono-cytokine-secreting counterpart. *Figure 5* shows three exemplary IL-6+/TNF-α+ secreting cells in response to 1 hr LPS stimulation identified after a manual analysis of 50 randomly selected CSCs. We found mainly three distinct dynamic patterns in these cells. The patterns included simultaneous secretion of both cytokines (*Figure 5A*, roughly 40%, i.e. 20 of 50 cells), sequential secretion behavior (where one cytokine ceased secretion before the other started, *Figure 5B*, roughly 40%), and semi-sequential secretion with overlap (*Figure 5C*, 20%).

To examine the overall change in secretion dynamics in polyfunctional cells and their mono-secreting counterparts over different stimulants and incubation times, the average in-droplet concentrations of

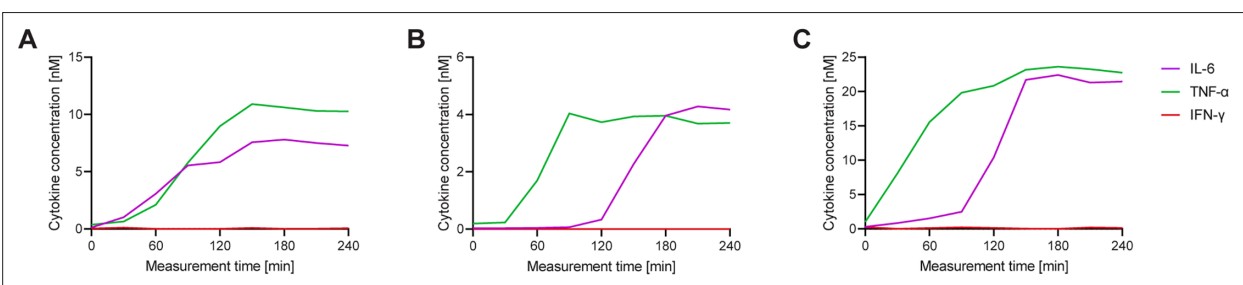

**Figure 5.** Each cell displays a different secretion behavior after LPS stimulation for IL-6+/TNF-α+ polyfunctional cells. Three exemplary cells secreting IL-6+/TNF-α+ after a 1 hr stimulation with 1 µg/ml LPS and subsequent single-cell measurement for 4 hr (**A–C**) Measured cytokines included IL-6 (purple), TNF-α (green), and IFN-γ (red). Secretion dynamics are visualized by plotting measured in-droplet concentration against measurement time.

the whole secreting population, as well as the average secretion rate of each individual cell were extracted over the measurement time (*Figure 6*).

The frequency of IL-6$^+$/TNF-$\alpha^+$ CSCs showed the biggest change after LPS and PHA stimulations, so we decided to analyze the dynamics of these cells with greater detail first. IL-6 secretion dynamics were similar for polyfunctional and mono-secreting cells in response to LPS stimulation (*Figure 6A*). The polyfunctional cells showed slightly higher endpoint concentrations with both short (6.7 vs. 5.0 nM) and long stimulations (10.4 vs. 8.5 nM). However, there were no significant changes in the secretion rate distributions of polyfunctional cells (p=0.36). On the other hand, average TNF-$\alpha$ endpoint concentrations of polyfunctional cells were increased compared to mono-cytokine-secreting cells (57.6 vs. 34.4 nM), with this difference disappearing upon prolonged stimulation (13.3 vs. 14.1 nM). A similar behavior for TNF-$\alpha$ secretion was found in the same subpopulation when the PBMCs were stimulated with PHA (1 hr polyfunctional 39.0 vs. TNF-$\alpha$ only 9.7 nM, *Figure 6B*). In contrast to LPS-stimulated cells, increased average in-droplet concentrations for polyfunctional cells were still observed after 24 hr (48.1 vs. 10.7 nM). This could potentially indicate a longer-lasting stimulation effect of PHA compared to LPS. However, while polyfunctional cells reached slightly higher endpoint concentrations after 1 hr for IL-6 compared to mono-IL-6 secretors (7.3 vs. 2.4 nM), this observation was inverted after 24 hr of stimulation (4.4 vs. 9.4 nM). Comparing the secretion rate distributions of IL-6$^+$/TNF-$\alpha^+$ polyfunctional cells in relation to prolonged stimulation, a significant difference in TNF-$\alpha$ secretion rate distribution was observed between 1- and 24 hr stimulations (p<0.0001), but not for IL-6 secretion rates.

Next, we investigated the response of TNF-$\alpha^+$/IFN-$\gamma^+$ polyfunctional cells where we observed a decrease in response to PMA/ionomycin, but an increase for anti-CD3/anti-CD28 stimulation when comparing 1- and 24 hr stimulations (*Figure 2C/D*). After stimulation with anti-CD3/anti-CD28 antibodies, secreting cells showed a similar dynamic profile for short and long stimulation times (*Figure 6C*). Polyfunctional cells displayed increased IFN-$\gamma$ in-droplet concentrations after 1- and 24 hr stimulations, compared to mono IFN-$\gamma$ secreting cells (16.4 vs. 7.2 nM and 25.4 vs. 14.4 nM, for 1 and 24 hr, respectively). A similar behavior was observed for the average TNF-$\alpha$ concentrations of polyfunctional cells, with higher concentrations reached for after 1- and 24 hr stimulations, compared to mono-TNF-$\alpha$ secretors (18.6 vs. 11.4 nM and 34.4 vs. 18.6 nM, respectively). Comparing secretion rate distributions of polyfunctional cells and their change over time, there were significant differences in the secretion rate distributions of polyfunctional cells for IFN-$\gamma$ and TNF-$\alpha$ (p=0.0024 and p=0.0005). For both cytokines, an additional high cytokine-secreting population (>100 molecules/s) appeared at prolonged incubation times. PMA/ionomycin stimulated PBMCs showed a similar behavior, with increased average IFN-$\gamma$ in-droplet concentrations for polyfunctional cells compared to their mono-IFN-$\gamma$ secreting cells (37.8 vs. 15.3 nM and 39.6 vs. 25.0 nM for 1- and 24 hr stimulations, respectively). Interestingly, almost the same average endpoint concentration was reached for both stimulation times, which was not the case after anti-CD3/anti-CD28 stimulation (*Figure 6D*). TNF-$\alpha$ on the other hand displayed very similar secretion dynamics, with the exception of mono-TNF-$\alpha$ secreting cells after 1 hr of stimulation. Interestingly, significant differences in secretion rate distributions were observed for both cytokines, comparing the polyfunctional cell populations over the stimulation time (p<0.0001 for both). IFN-$\gamma$ secreting cells displayed a shift towards higher secretion with prolonged stimulation, with 18.4% of the cells exceeding >1'000 molecules/s after 24 hr, compared to 0.3% after 1 hr.

Finally, we wanted to analyze the response of IL-8$^+$/MIP-1$\alpha^+$ polyfunctional cells after stimulation (*Figure 6E/F*). For zymosan-stimulated cells, similar secretion dynamics were observed after 24 hr of stimulation for both IL-8 and MIP-1$\alpha$ (*Figure 6E*). Interestingly, more distinct differences were apparent after the 1 hr stimulations, with average IL-8 concentrations greatly increased after 20 min for polyfunctional cells, compared to mono-IL-8 secreting cells, leading to vastly different endpoint concentrations (42.8 vs 13.5 nM). Similar behavior was observed for MIP-1$\alpha$ secreting cells, where average in-droplet concentrations showed no difference up to 60 min into the measurement. After this time point, the polyfunctional cells started to secrete more MIP-1$\alpha$, resulting in a higher endpoint concentration at the end of the 4 hr measurement (9.1 vs. 2.4 nM). Even though the differences in concentrations were more pronounced early, the polyfunctional cells always reached slightly higher endpoint concentrations of both cytokines after 24 hr stimulations (23.7 vs. 14.5 nM and 36.4 vs. 30.7 nM for IL-8 and MIP-1$\alpha$, respectively). Interestingly, while overall IL-8 secretion of polyfunctional

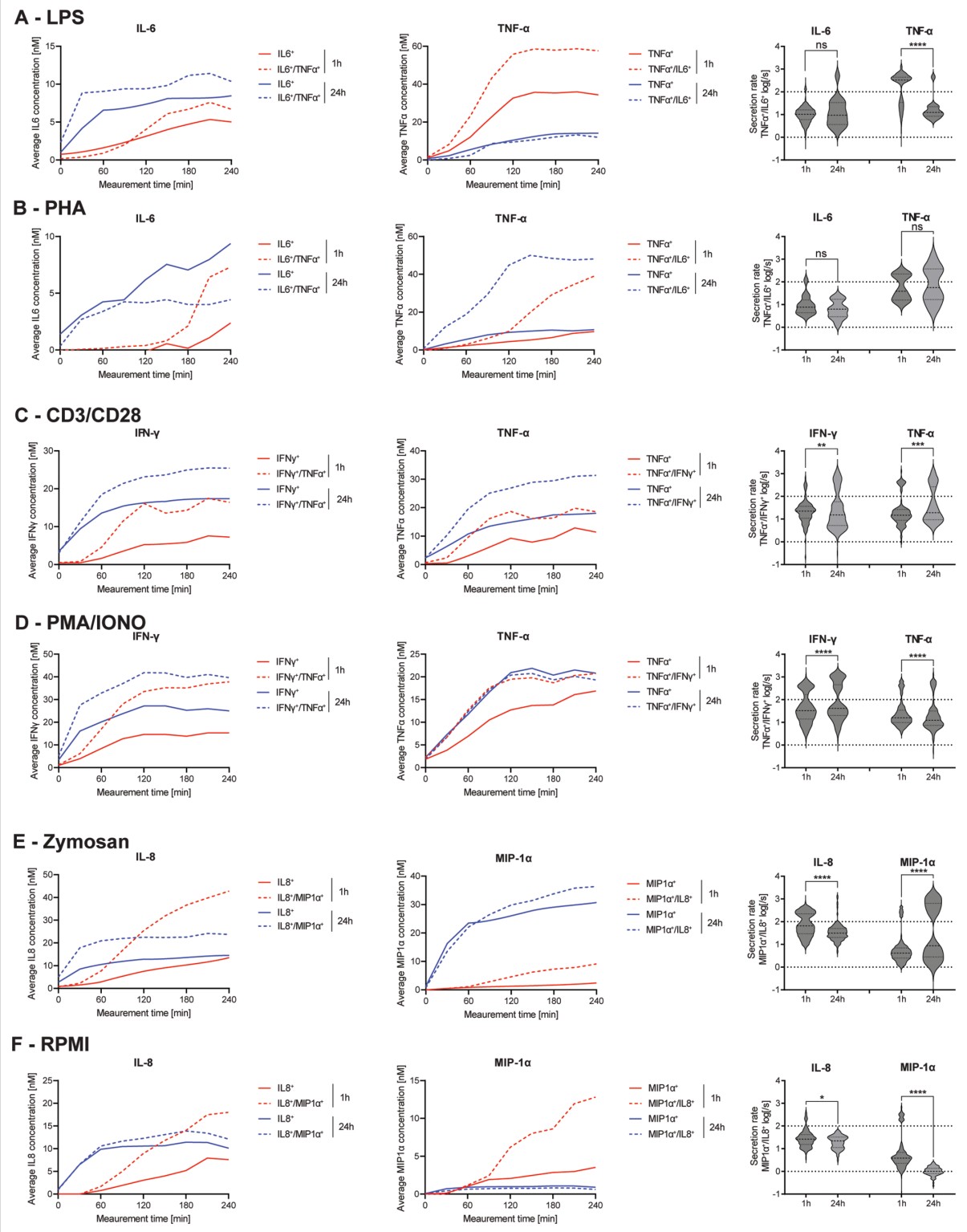

**Figure 6.** Secretion dynamics change for polyfunctional cytokine-secreting populations. Secretion behavior changes for co-secreting cells with prolonged incubation times. IL-6⁺/TNF-α⁺ secreting cells in response to 1- and 24 hr stimulations with LPS (**A**) and phytohemagglutinin (PHA) (**B**). TNF-α⁺/IFN-γ⁺ secreting cells in response to 1- and 24 hr stimulations with anti-CD3/anti-CD28 (**C**) and PMA/ionomycin (**D**). IL-8⁺/MIP-1α⁺ secreting cells in response to 1- and 24 hr stimulations with zymosan (**E**) and for non-stimulated cells (**F**). First¹ and second panels show the average in-droplet concentrations of the respective populations over the measurement time (4 hr), compared to the single-cytokine secretion population. Third panels show the average secretion rate (log) distributions of individual IL-6⁺/TNF-α⁺ secreting cells over the whole measurement time for 1- and

*Figure 6 continued on next page*

*Figure 6 continued*

24 hr stimulations and for both measured cytokines. Statistical differences in secretion rate distributions were assessed using two-sided, unpaired, nonparametric Kolmogorov-Smirnov t-tests with 95% confidence and p<0.05. Data with less than ten detected secreting cells is not depicted. Vertical dotted lines show secretion rates of 1 and 100 molecules per second, respectively. $n_{biological\ replicates}$=1.

cells decreased with prolonged stimulation, it greatly increased for MIP-1α, also observable in the average secretion rate distributions of the single-cells. Significant differences were visible over time for polyfunctional cells (p<0.0001 for both), with IL-8 seeing a reduction of cells secreting >100 molecules/s (39.9% vs. 4.2%). On the other hand, MIP-1α polyfunctional cells, showed a great increase for these high-secreting cells (8.9% vs. 43.7%). Non-stimulated PBMCs showed similar behavior to zymosan-stimulated PBMCs for IL-8 secretion (*Figure 6F*). In short, an increased average concentration for polyfunctional cells compared to mono-IL-8 secreting cells early (18.0 vs. 7.6 nM) and late was measured (12.1 vs. 10.1 nM). Unlike zymosan-stimulated cells, MIP-1α concentrations were greatly increased for polyfunctional cells after the 1 hr incubation (12.8 vs. 3.5 nM), with this difference diminishing after 24 hr of incubation (0.6 vs. 0.9 nM). Secretion rate distributions were only slightly significantly different over the stimulation time for IL-8 polyfunctional cells (p=0.03), mainly due to some high-secreting cells disappearing after 24 hr. In contrast, MIP-1α showed large differences in the secretion rate distribution of mono-secreting cells (p<0.0001) with an overall shift towards lower secretion.

In summary, the detailed analysis of the subpopulation revealed a general trend to higher secretion rates and secreted concentrations in polyfunctional cells.

## Discussion

Cytokine polyfunctionality is a well-described concept in T cells and has been associated with immunological disease control in the human body (*Boyd et al., 2015*; *Casazza et al., 2006*; *Imai et al., 2020*; *Betts et al., 2006*). In this study, we aimed to investigate the polyfunctional cytokine secretion response of PBMCs from healthy donors stimulated with a variety of well-known stimulants with greater dynamic sensitivity and granularity, and to distinguish between simultaneous and sequential cytokine secretion. Indeed, the observed changes in polyfunctionality were ambiguous and complex, with a general trend towards reduced polyfunctionality with prolonged stimulation times in terms of activated cell numbers (*Figure 2*, *Figure 3*). It is also important to highlight that the observed reduction in polyfunctional cells did not result in an increase of mono-cytokine-secreting cells. This indicated a reduction of overall polyfunctionality with prolonged stimulation and not only the ceasing of the secretion of one cytokine, an observation is also shown in other studies (*Han et al., 2012*). Interestingly, stimulations with anti-CD3/anti-CD28 and PHA displayed more polyfunctional cells after prolonged stimulations. This could be due to the slower mechanism of action of those stimulants, mainly acting on T cells (*Movafagh et al., 2011*; *Ai et al., 2013*; *Olsen and Sollid, 2013*). The frequency of polyfunctional cells could, therefore, be indicative of the current immune state, as it has been proposed before for T cell efficacy during infection (*Boyd et al., 2015*). Additionally, a general reduction of polyfunctional cells and shift towards mono-cytokine secretion has been shown before in response to potent stimulants, such as PMA/ionomycin (*Han et al., 2012*).

Interestingly, we also observed short-lived polyfunctional cells when using innate immune cell stimulants such as LPS and zymosan (*Lu et al., 2008*; *Dillon et al., 2006*). Cytokine polyfunctionality is not well described for innate immune cells. However, this could indicate that a decrease in polyfunctionality happens across different immune cell populations, and the state of polyfunctionality is very short-lived and regulated in innate immune cells, potentially a sign of a more specialized, granular, and directed response. However, additional studies are needed to fully decipher and understand the dynamics and impact of polyfunctional innate immune cells.

We did not observe a general correlation between the frequency of polyfunctional cells and the concentration of secreted cytokines, since different responses were observed depending on the measured cytokine and used stimulant, exemplarily seen by the differences for TNF-α⁺/IFN-γ⁺ secreting cells in response to anti-CD3/anti-CD28 and PMA/ionomycin stimulations. While there was a decrease in the frequency of polyfunctional cells after prolonged PMA/ionomycin stimulation, the data displayed an increase for anti-CD3/anti-CD28 stimulated cells (*Figure 2C/D*). But notably, both

cell populations exhibited a substantial increase in the secretion rates of IFN-γ and TNF-α over the measurement time (*Figure 4E*).

Differences also occurred within the same population of polyfunctional cells, by upregulation of one and downregulation of the other cytokine. This was observed for LPS-stimulated PBMCs, where prolonged stimulation led to a reduction of IL-6$^+$/TNF-α$^+$ secreting cell numbers, decreased TNF-α, but increased IL-6 secretion rates. Again, this could indicate a granular shift in the response to a more specialized answer, upregulating needed cytokines, downregulating unneeded cytokines, and reducing the number of secreting cells. Supporting this hypothesis is the fact that there is a clear trend towards higher secretion rates in polyfunctional cells, compared to their mono-cytokine counterparts. This also underlines the importance of those cell populations in a directed and efficient immune response.

While a trend towards higher secretion rates was observed, no general trend in distinct secretion patterns differentiating mono secretors from polyfunctional cells was identified (*Figure 6*). For the majority of stimulations, similar secretion dynamics between mono- and polyfunctional cells were observed. However, polyfunctional cells consistently displayed elevated in-droplet concentrations, confirming their similar secretion dynamics but higher secretion levels compared to the mono-cytokine-secreting counterpart. Yet, some notable exceptions to this trend were found such as for TNF-α secretion in response to LPS. A distinct lag phase was found for polyfunctional cells compared to mono-TNF-α secretors early after stimulation that was absent at prolonged incubation, potentially indicating a still mounting immune response distinct to polyfunctional cells.

Potential limitations of the herein performed study and its system lies in the low parallelization of cytokine analysis. While we confirmed the presence of the six cytokines in bulk, we only measured three cytokines in parallel, and in a limited amount of combinations. Combinations of cytokines were grouped according to expected secretion behavior to observe overlaps between different cell types within the PBMC population, i.e., to increase the chance of observing polyfunctional cytokine secretion in both panels. Nevertheless, some polyfunctionality is masked as not all combinations were assayed. Indeed, we observed mainly three populations of polyfunctional cells: IL-6$^+$/TNF-α$^+$, TNF-α$^+$/IFN-γ$^+$, and IL-8$^+$/MIP-1α$^+$. In addition, single-cell measurement methods also inherently limit paracrine cell signaling and stimulation throughout the measurement, although some paracrine stimulation could occur in our experiments due to the incubation in bulk. We decided to use PBMCs in this study, with the benefit of having a higher clinical relevance for future applications in patient stratification and monitoring. However, due to this choice, we cannot make assumptions about secretion behavior of individual cellular populations. There is also a valid argument for preselecting target populations, such as T cells, dendritic cells, or subpopulations, in order to obtain an isolated, more biologically focused perspective on the cellular subset, and establish correlations between the observed responses and the specific cell types involved. An interesting approach would be the isolated view on innate immune cells, which we included but not isolated in our measurements, due to their polyfunctionality being linked with age-related functional and cognitive decline in elderly adults (*de Pablo-Bernal et al., 2016*). Indeed, preselecting cellular subpopulations might also increase the number of secreting cells per measurement, further increasing the statistical power of the method. Nevertheless, the enrichment process, whether conducted before or after stimulation, could impact the secretion patterns of specific cells, as well as reducing clinical significance if only one subpopulation is stimulated in isolation.

In summary, polyfunctional cytokine responses are equivocal and complex, but could offer a valid readout for the dynamic nature of the immune system. Additionally, the here deployed assays offer in-depth characterization and a multiplexed readout of individual CSCs overtime. The observed 'fingerprints' and biological signatures could be used to assess the immune status in health and disease and facilitate the use of personalized treatment in immune-related disorders (*Ghobadine-zhad et al., 2022*). This could be used to answer fundamental research questions, understand disease pathology and in the clinical monitoring of patients.

# Materials and methods

## Cell handling and stimulation

The herein analyzed measurements were part of a study published elsewhere (*Portmann et al., 2023*). In brief, peripheral blood mononuclear cells were isolated from buffy coat and stored in liquid nitrogen until use. For stimulation experiments, the cells were thawed, stained with CellTrace violet, and treated with human FcR blocking reagent. After washing and counting, the cells were diluted and stimulated with 1 µg/ml lipopolysaccharide (Invivogen), 50 ng/ml Phorbol-myristate-acetate, and 1 µg/ml Ionomycin (both Sigma-Aldrich), 100 µg/ml zymosan (Sigma-Aldrich), 10 µg/ml phyto-hemagglutinin-L (Thermo Fisher), or 5 µg/ml anti-CD3 (OKT3, Thermo Fisher, #16-0037-85; RRID: AB_468855), and anti-CD28 (CD28.2, Thermo Fisher, #16-0289-85; RRID: AB_468927) for 1 and 24 hr using ultra-low binding plates and cell concentrations of $10^6$ cells/ml. All experiments were performed using PBMCs from anonymized, healthy donors and were carried out under ethics agreement EK202-N-56, approved by the ETH Zurich ethics commission. Due to the anonymized samples, no information about sex, age, and gender was available.

## Cell encapsulation and microfluidic measurements

Immediately before encapsulation, cells were washed to avoid cross-contamination and diluted to concentrations averaging 0.2–0.4 cells/droplet. Cells were encapsulated together with assay reagents, including functionalized magnetic nanoparticles (AdemTech) and corresponding fluorescent detection antibodies. Panel 1 included a mixture of nanoparticles functionalized against IL-6, TNF-α, IFN-γ, panel 2 a mixture for IL-2, IL-8, MIP-1α, with the respective cytokines measured on the different fluorescence channels. The following antibodies were used: IL-6 Monoclonal Antibody Biotin (MQ2-39C3, Thermo Fisher, #13-7068-85, AB_466913), IL-6 Monoclonal Antibody FITC (MQ2-13A5, Thermo Fisher, #11-7069-81, AB_465396), TNF-alpha Monoclonal Antibody (biotinylated in-house, MAb1, Thermo Fisher, #14-7348-85, AB_468488), TNF alpha Monoclonal Antibody PE (MAb11, Thermo Fisher, #12-7349-81, AB_466207), IFN gamma Monoclonal Antibody (biotinylated in-house, MD-1, Thermo Fisher, #14-7317-85, AB_468474), IFN gamma Monoclonal Antibody APC (4 S.B3, Thermo Fisher, #17-7319-82, AB_468477), Goat Anti-Human IL-2 Polyclonal antibody Biotin (R&D Systems, #BAF202, AB_356218), IL-2 Monoclonal Antibody FITC (MQ1-17H12, Thermo Fisher, #11-7029-42, AB_2572512), Rabbit Anti-Human IL-8 (biotinylated in-house, PeproTech, #500-P28, AB_147577), Mouse Anti-Human IL-8 Monoclonal Antibody (AF555 labeled in-house, PeproTech, #500-M08, AB_147487), Goat Anti-Human MIP-1a Polyclonal Antibody Biotin (PeproTech, #500-P38BT, AB_147602), CCL3 Polyclonal Antibody (AF647 labelled in-house, Thermo Fisher, #PA5-47000, AB_2609538). Please refer to *Portmann et al., 2023* for detailed information about the nanoparticle functionalization, used reagents, and concentrations. Cell and assay reagent encapsulation into 65 picoliter water-in-oil droplets and subsequent immobilization thereof was performed as described by *Bounab et al., 2020* Imaging was performed using a Nikon TI2 Eclipse epifluorescence microscope with fluorescence measurements every 30 min up to 4 hr. The sample was maintained in a darkened cage incubator at 37 °C and imaged using a 10 x objective.

## Data analysis

Image and data analysis was performed using a custom Matlab script (Mathworks, version R2020A) available for download on GitHub (*Chenon, 2022*; https://github.com/ESPCI-LCMD/MiMB) as described by *Portmann et al., 2023*. In brief, cells were identified as secreting cells by the droplets meeting four criteria: Positive identification of the CellTrace stain (DAPI channel), fluorescence relocation onto the magnetic nanoparticles exceeding $\mu_{Relocation}0 + 1.645 * 2 * \sigma_{BLK}$ over the measurement time, the change between maximum and minimum beadline relocation is greater than $1.645 * 2 * \sigma_{BLK}$ and the fluorescence relocation increases over the measurement time (slope >0). Polyfunctional cells were identified by meeting the above-mentioned criteria for more than one cytokine, i.e. more than one fluorescence channel. Relocation was quantified into concentration using calibration curves as described by *Portmann et al., 2023* If the number of droplets identified as positive according to the criteria was fewer than 50, manual inspection was performed. If fewer than 10 droplets met the visual inspection criteria (absence of fluorescence aggregates and presence of intact beadline), they were excluded from the analysis. To calculate secretion rates, the concentration change between consecutive time points was converted into molecules per second, and these values were averaged across all

time points. If the maximum detectable concentration was reached before the end of the experiment, the concentration was set to the maximum the secretion rate was averaged only until this time point. Percentage in bin for *Figure 3A* was calculated according to the following formula:

$$\%_{Bin} = \frac{\sum n_{cellsinbin}}{\sum n_{cellspermeasurement}}.$$

The generated raw data that were analyzed during this study are openly available in the Dryad repository (*Portmann et al., 2024*).

## Cytokine supernatant measurements

Cytokine concentration in supernatants was measured using the CodePlex secretome analysis platform from Isoplexis. The human cytokine storm panel measures 18 cytokines, including IL-2, IL6, IL-8, TNF-α, IFN-γ, and MIP-1α. Experiments were performed according to the supplier's instructions on an Isospark device (Isoplexis). Data analysis was performed using the proprietary IsoSpeak software (2.9.0.811, IsoPlexis).

## Statistical tests and biological replicates

Unless specified, replicates are represented as mean ± SEM. Statistical differences between mono- and polyfunctional cell populations were assessed using paired two-sided t-tests with a 95% confidence level (*Figure 3*). Differences in distributions were evaluated using two-sided, unpaired, nonparametric Kolmogorov-Smirnov t-tests with a 95% confidence level. p-values were indicated as follows: * (0.05–0.01), ** (0.01–0.001), *** (0.001–0.0001), and **** (<0.0001). n represents the number of biological replicates.

## Acknowledgements

This project was supported by the grant #2021–349 of the Strategic Focus Area 'Personalized Health and Related Technologies (PHRT)' of the ETH Domain (Swiss Federal Institutes of Technology), the Swiss National Foundation grant #310030_197619, the Olga-Mayenfisch Stiftung, and the Novartis Foundation for medical-biological research.

## Additional information

### Funding

| Funder | Grant reference number | Author |
| --- | --- | --- |
| Personalized Health and Related Technologies | #2021-349 | Klaus Eyer |
| Swiss National Science Foundation | #310030_197619 | Klaus Eyer |
| Olga Mayenfisch Stiftung | | Klaus Eyer |
| Novartis Foundation for Medical-Biological Research | | Klaus Eyer |

The funders had no role in study design, data collection and interpretation, or the decision to submit the work for publication.

### Author contributions

Kevin Portmann, Conceptualization, Data curation, Formal analysis, Investigation, Visualization, Methodology, Writing - original draft, Writing – review and editing; Aline Linder, Investigation, Methodology, Writing – review and editing; Klaus Eyer, Conceptualization, Supervision, Funding acquisition, Methodology, Writing – review and editing

## Author ORCIDs
Kevin Portmann ⓘ http://orcid.org/0009-0008-3690-5254
Aline Linder ⓘ http://orcid.org/0009-0008-0965-1949
Klaus Eyer ⓘ https://orcid.org/0000-0001-9344-5110

## Ethics

All experiments were performed using PBMCs from anonymized, healthy donors supplied by the bloodbank in Zurich and were carried out under ethics agreement EK202-N-56, approved by the ETH Zurich ethics commission. Due to the anonymized samples, no information about sex, age, and gender was available.

Reviewer #1 (Public review): https://doi.org/10.7554/eLife.89781.3.sa1
Reviewer #2 (Public review): https://doi.org/10.7554/eLife.89781.3.sa2
Author response https://doi.org/10.7554/eLife.89781.3.sa3

---

## Additional files

### Supplementary files
• MDAR checklist

### Data availability

The generated raw data that was analyzed during this study are openly available in the Dryad repository, https://doi.org/10.5061/dryad.612jm64c2.

The following dataset was generated:

| Author(s) | Year | Dataset title | Dataset URL | Database and Identifier |
|---|---|---|---|---|
| Eyer K | 2024 | Data from: Stimulation-induced cytokine polyfunctionality as a dynamic concept | http://dx.doi.org/10.5061/dryad.612jm64c2 | Dryad Digital Repository, 10.5061/dryad.612jm64c2 |

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
