## [Editor Report · eLife assessment]

This **useful** study uses a microfluidic method to evaluate the ability of single human white blood cells to produce combinations of cytokines and the evidence that this takes place is **solid**. The paper highlights polyfunctionality using data that are similar to a prior dataset from the same group. The authors comment that, in analysis of larger panels, single cells rarely make more than 2 or 3 cytokines so that investigation of 3 cytokines at a time is sufficient to investigate this phenomenon. Coupling this approach to other modes of single cell analysis may provide greater insight into what limits simultaneous production of multiple cytokines.

---

## [Referee Report · Reviewer #1 (Public review)]

Summary: The authors started by stimulating the PBMCs in bulk, then encapsulated single cells in droplets to monitor the secreted cytokines in each droplet for the next 4 hours. The secreted cytokines are bound by fluorescently labeled detection antibodies. At the same time, the cytokines can be captured by the capture antibodies that are immobilized to the magnetic beads. Under the magnetic field, the magnetic beads will line up in the middle of the droplet along with bound fluorescent antibodies. This effectively enriches the fluorescent antibody to the middle of the droplet, making it a higher fluorescent signal compared to the background signal that is in the rest of the droplet. They can parallel the measurement of three cytokines in each droplet.

Strengths: Observed heterogeneous cytokine secretion dynamics, which they have reported in their previous paper as well.

Weaknesses:

Since they used PBMCs, without other assay to confirm the cell subtypes, I am not sure if any of the heterogeneity they detected in 6 cytokine secretion would be able to relate back to biology. In addition, the two panels were measured on separate cells, I am not sure it is meaningful to make any comparisons of the two panels as they are on different cells.

Their revision failed to make much improvement.

---

## [Referee Report · Reviewer #2 (Public review)]

The responses to the comments and changes in the manuscript are convincing, especially the secretion patterns of high and low secreting cells are interesting and reassuring. The only criticism I still have is that most observations are already published in the previous paper by the same authors.

Summary:

In their manuscript titled "Stimulation-induced cytokine polyfunctionality as a dynamic concept," the authors investigate the dynamic nature of polyfunctional cytokine responses to established stimulants. The authors use their previously published single-cell encapsulation droplet-microfluidic platform to analyse the response of peripheral blood mononuclear cells (PBMCs) to different stimulants and measure the secretion dynamics of individual cytokines. This assay shows that polyfunctionality in cytokine responses is a complex but short-lived phenomenon that decreases with prolonged stimulation times. The study finds that polyfunctional cells predominantly display elevated cytokine concentrations with similar secretion patterns but higher secretion levels compared to their monocytokine-secreting counterparts. The method is promising to analyse the correlation between the secretion dynamics of different cytokines in primary samples and heterogeneous cell populations.

Strengths:

This method provides single-cell-resolved and dynamic cytokine concentration information, which might be used to identify "fingerprints" of secretion patterns for selected cytokines. When extending the available data to more than one donor, this might be the basis for a diagnostic tool. The combination of established droplet microfluidics with an epi-fluorescence microscope-based readout makes it convincing that the method is transferable to other labs. Specifically, the dynamic analysis of cytokine concentrations is interesting, and the differences or similarities in secretion timepoints might be missed with end-point methods. The authors convincingly show that they detect up to three different cytokines in single cells.

Weaknesses:

The conclusions of the study are based on samples from a single donor, which makes the conclusions on secretion patterns difficult to interpret. The choice of cytokines is explained, but the justification of the groupings of the antibodies into the two panels is missing. It would further be helpful to discuss how the single cell incubation might affect the secretion dynamics vs. the influence of co-culture of all cell types during the 24 h activation. The authors compare average secretion rates and levels. However, the right panel in Fig. 6 looks like there might be two different populations of mono- or polyfuntional cells that have two secretion rates. As the authors have single-cell data, I would find the separation into these populations more meaningful than comparing the mean values. In line with this comment, comparing the mean values for these cytokines instead of the mean of the populations with distinct secretion properties might actually show stronger differences than the authors report here.Is the plateau of the cytokine concentration caused by the fluorescence signal saturating the camera, saturation of the magnetic beads, exhaustion of the fluorescent antibodies, or constant cytokine concentrations? The high number of non-CSCs and the limited number of droplets decrease the statistical power of the method. The authors discuss their choice to use PBMCs and not solely T cells, but this aspect is missing in the discussion.

---

## [Author Response]

The following is the authors’ response to the original reviews.

**Reviewer 1**
Since they used PBMCs, without other assays to confirm the cell subtypes, I am not sure if any of the heterogeneity they detected in 6 cytokine secretion would be able to relate back to biology.

We agree with the reviewer that we cannot relate cytokine secretion back to specific cell populations and that part of the heterogeneity observed is due to various cellular populations and subpopulations. However, we would argue that the results obtained from measuring PBMCs especially relate to biology, not cellular identity, and provide useful information on how PBMCs will respond to a specific challenge since they offer more clinical relevance in patient stratification and monitoring. Thus, the possibility of identifying trends in polyfunctional cytokine secretion is not hindered by the isolated view of one specific cellular subpopulation. However, we agree that future experiments must identify the polyfunctional cells and decipher the extent of heterogeneity within the population.

In addition, the two panels were measured on separate cells, I am not sure it is meaningful to make any comparisons of the two panels as they are on different cells.

Thank you for mentioning this point. If this refers to Figure 3, where we compare the percentage of secreting cells incubation times, these cells are all individual data points, i.e., individual cells and then pooled. It is true that, potentially, these could be similar cell types (a cell co-secreting TNFa/IL-6 could also co-secrete IL-8/MIP-1a). Since they originate from the same cell batch and stimulation, only divided before encapsulation, we think it is a valid comparison as this would also be done in ELISpot or similar techniques.

**Reviewer 2**
The conclusions of the study are based on samples from a single donor, which makes the conclusions on secretion patterns difficult to interpret. The choice of cytokines is explained, but the justification of the groupings of the antibodies into the two panels is missing.

Thank you for highlighting this valid criticism. We chose to use cells from one donor to examine the secretion patterns observed in one individual, as cells from different individuals might respond differently. The focus of the experiments described in this study was to describe secretion patterns with respect to the incubation times and secreted cytokine, including multiple donors, which would address a different question (i.e., how is polyfunctionality different between individuals). The cytokines were grouped according to expected secretion to observe overlaps between different cell types (to increase the chance of seeing secretion from both panels simultaneously). We have added complementary text discussing the justification of cytokine grouping in the updated manuscript.

It would further be helpful to discuss how the single cell incubation might affect the secretion dynamics vs. the influence of co-culture of all cell types during the 24 h activation.

Thank you for this input. We discussed this potential limitation in detail in a previous publication (Portmann et al., Cell Reports Methods, 2023) and added some addressing sentences to the discussion.

The authors compare average secretion rates and levels. However, the right panel in Fig. 6 looks like there might be two different populations of mono- or polyfuntional cells that have two secretion rates. As the authors have single-cell data, I would find the separation into these populations more meaningful than comparing the mean values. In line with this comment, comparing the mean values for these cytokines instead of the mean of the populations with distinct seretion properties might actually show stronger differences than the authors report here.

Thank you for this addition. This plot focuses on describing the relationship between secretion and incubation times. We agree that the data can be further divided into high and low secretion and the respective average plot. However, we finally decided against such a solution to avoid bias due to small event counts in certain high- and low-polysecreting populations. We checked whether dynamics are different between these populations, and the individual averages largely follow the overall trend, although on different plateaus – indeed, high-secreting cells will reach a plateau due to saturation. We have added the plot for IFNy here to visualize this point.

Is the plateau of the cytokine concentration caused by the fluorescence signal saturating the camera, saturation of the magnetic beads, exhaustion of the fluorescent antibodies, or constant cytokine concentrations?

Thank you for raising this point. On the individual cell level, the plateau is caused by assay capacity limitations for high-secreting cell populations, i.e., the capacity of the nanoparticles. For low secreting populations, the plateau is caused by a cease in secretion, whereas for high-secreting cells, the capacity will be limiting. This has been extensively discussed in Portmann et al., Cell Report Methods, 2023.

The high number of non-CSCs and the limited number of droplets decrease the statistical power of the method. The authors discuss their choice to use PBMCs and not solely T cells, but this aspect is missing in the discussion.

As mentioned above, we chose PBMCs for their better representability and heterogeneity in clinical settings. Indeed, focusing on secreting cell subpopulations would increase the percentage of CSCs and the number, but we found the method to be sufficiently statistically powerful for our measurements. However, we also agree with the comment raised by reviewer 1 that a focus on a specific cell population might be interesting for many questions and applications. We have added respective text to the discussion section.

The absolute cell number is missing. This might also answer the question of whether polyfunctional cells turn into monofunctional cells after stimulation for 24 hours or if the monofunctional population expands more.

We are unsure of this comment. If the reviewer refers to a potential expansion ex vivo over 24 h, we have checked this for different conditions and could not observe cellular expansion within this timeframe – the numbers remained mostly stable, sometimes decreasing and only increasing in CD3/CD28. However, an overall change in cell counts does not necessarily relate to the functionalities of individual cells. This observation, combined with our results, hints towards a dynamic cellular restriction of polyfunctionality, but is no direct evidence for such a hypothesis as individual cells need to be followed in such an experiment over a much larger time frame.

Fig. 4: Using a divergent colour scheme would be helpful. Fig. 6: Adding labels with the stimulation next to the plots would be helpful.

We have changed the figures accordingly.

A limitation of the approach is that the detection of polyfunctionality relies on how the three cytokines in each panel are selected and comparisons between the two panels are not otherwise helpful. Can the authors discuss how many panels would be needed to fully explore polyfunctionality among the six cytokines?

Thank you for this comment. We agree that the identification of polyfunctional cells is dependent on the panel selection, and its composition. We had to select respective panels, and based our initial choice for this study on expected secretion behavior from PBMCs, instead of engineering panels specific for one cell type. However, these panels can be adapted to study additional questions. Interesting point. 6 cytokines into groups of 3 allows for 20 possible combinations. However, we very rarely see triple positive polyfunctional cells, and not all combinations would make sense due to cellular restrictions and differences in stimulations.

Is there any way to increase the number of cytokines that could be detected in one droplet?

This can be done on a lower throughput scale by removing the Cell Trace violet stain. This would allow the current method to measure up to 4 cytokines. An alternative would be adding different fluorophores without spectral overlap so that the throughput could increase to around 6-7 max, allowing us to measure polyfunctionality in a less biased manner. Other solutions are needed if >6-7 cytokines should be measured. Our experiments (with high-throughput cytokine detection systems, Fireplex and Isoplexis, i.e., 17-18 cytokines) showed that cells rarely secreted more than three cytokines at a time.